# OpenReview forum: "Do LLMs Forget What They Should? Evaluating In-Context Forgetting in Large Language Models"
_ICLR.cc/2026/Conference — ICLR 2026 Poster_

### Official Review · Reviewer_XYBR · 2025-10-22

**Soundness:** 3
**Presentation:** 2
**Contribution:** 3
**Rating:** 6
**Confidence:** 4

**Summary:**

The paper proposes ICF-Bench, a benchmark that measures Large Language Models (LLMs)' ability to follow instructions while taking into account explicit or implicit updates, called forget interference, in the conversation. The dataset was constructed by inserting LLM-generated forget interferences into existing multi-turn dialogue datasets. Various open-weight and closed models are evaluated on the benchmark, consisting of 3 scenarios (instructional forgetting, subtask revision, and dynamic preference), using 3 metrics  (NoForget Accuracy, Forget Accuracy, and SFRR). The performance of the models degrades when forget interferences are introduced. SFRR does not always increase with model size. When increasing the context size, the models behave differently depending on the scenario and the metric.

**Strengths:**

- The paper proposes a simple yet informative benchmark. It could help the community correctly assess the current limitations of LLMs.
- The paper provides a lot of details about dataset creation and model evaluation. This can make replication easier.
- A wide range of models (open-weights/closed  and of different sizes) are evaluated on the proposed benchmark.

**Weaknesses:**

- The analysis part could be more extensive. The paper does not go deeper in explaining the causes of the different observed phenomena. It only proposes vague conjectures (like in line 146 and line 455). For example, recency bias , which was observed and analyzed in prior work, could explain SFRR score as a function of context length on the subtask revision scenario.
- The human evaluation is not very clear. Was it done only on one subject? Moreover, there is no statistical test on the agreement between human and automated evaluation.
- Concerning readability, Table 1 and 2 are a bit hard to parse. Turning them into a figure (at least the average scores) could improve the readability of the paper. Also, combining the Tables might help see the trend across the metrics and avoid repeating the NA scores across the tables. The x labels in Figure 3 are not aligned.
- Related work, you might find helpful the benchmark proposed in this paper https://arxiv.org/abs/2502.13791 that is very similar to this work (subtask revision and dynamic preference scenarios).

**Questions:**

1 - Some times SFRR improves with model size (For example, SFRR goes from 33% for Llama3-8B to 46% for Llama3-70B), what causes the variations in SFRR across models?
2 - Can the trends be described in a more quantitative way? what is the relation between the different metrics? scenarios?
3 - Could you provide more details about the human evaluation? To what extent is human evaluation in agreement with LLM evaluation?

---

> ### Author Response · Authors · 2025-11-21
> **Rebuttal (Page 1)**
>
> Thank you for your time and effort in reviewing our work. We would like to address the questions you raised as follows:
>
> ---
>
> >  The analysis part could be more extensive. The paper does not go deeper in explaining the causes of the different observed phenomena. It only proposes vague conjectures (like in line 146 and line 455). For example, recency bias , which was observed and analyzed in prior work, could explain SFRR score as a function of context length on the subtask revision scenario.
>
> Thank you for this insightful comment. We fully agree that understanding the underlying reasons for ICF performance issues is critical.
>
> Our contribution in the current paper is a comprehensive benchmark (ICF-Bench) and controlled paired evaluations to reveal capability gaps between memory and ICF. Nevertheless, we have begun to analyze the possible reasons behind the observed limitations in our paper:
>
> - "Instead, these findings suggest that in-context forgetting requires distinct cognitive mechanisms, potentially involving dynamic attention reallocation, conflict resolution between competing contextual signals, and robust suppression of outdated information, which are not automatically enhanced through scaling alone."(Chapter 4.3, Paragraph 1)
> - "This indicates a qualitative improvement in tracking and prioritizing evolving user intents, likely due to enhanced preference modeling in training."(Chapter 4.3, Paragraph 2)
> - "Overall, although long contexts generally exacerbate the challenges of in-context forgetting, they can mitigate the influence of early context in revision tasks, as the model’s limited robustness to long contexts sometimes leads to improved performance in ICF."(Chapter 5)
>
> Based on your suggestion, we will add some discussion on the underlying causes of model performance and expand Sections 4.2 and 4.4 to analyze these phenomena and their root mechanisms. For example:
> - After introducing forgetting interference, models generally perform worse compared to without forgetting interference. This may be because standard attention training objectives inherently prioritize retaining and leveraging historical information (favoring recall or continuation), and thus lack explicit “withdraw or delete” pathways. As a result, attention reallocation under conflict exhibits **attention inertia**[1], and different types of information (facts, subtasks, preferences) overlap in the vector space. This **representation entanglement**[2] makes simple natural language negation insufficient for the model to distinguish and discard specific components during retrieval, leading to a general inability to suppress or revoke prior content.
> - As context length increases, different scenarios exhibit divergent trends, likely due to the combined effects of **attention dilution**[3] and **recency bias**[4]. In dynamic preference, attention dilution weakens the relative strength of the latest preference, while recency bias becomes unstable with increasing distance, making the model more likely to revert to earlier preferences and causing performance degradation. In subtask revision, however, attention decay caused by distance reduces the influence of old subtasks, making it easier for the model to adopt the new instruction, thereby improving SFRR. The trend of instructional forgetting is similar to that of dynamic preference. These mechanisms collectively explain the directional differences in forgetting performance under long-context conditions.
>
> Finally, in Section 5, we will also propose exploring how to enhance LLMs' in-context capabilities as future work, including prompt engineering, attention-level modulation techniques, representation disentanglement, architectural forgetting gates，among other factors.
>
> [1] Hosseinzadeh R, Sadeghzadeh M. Attention Mechanisms in Transformers: A General Survey[J]. Journal of AI and Data Mining, 2025, 13(3): 359-368.
>
> [2] Wu G, Zhang S, Shi R, et al. Representation Entanglement for Generation: Training Diffusion Transformers Is Much Easier Than You Think[J]. arXiv preprint arXiv:2507.01467, 2025.
>
> [3] Zhang X, Chang X, Li M, et al. Selective attention: Enhancing transformer through principled context control[J]. Advances in Neural Information Processing Systems, 2024, 37: 11061-11086.
>
> [4] Clark C, Oh B D, Schuler W. Linear recency bias during training improves transformers’ fit to reading times[C]//Proceedings of the 31st international conference on computational linguistics. 2025: 7735-7747.

---

> ### Author Response · Authors · 2025-11-21
> **Rebuttal (Page 2)**
>
> > The human evaluation is not very clear. Was it done only on one subject? Moreover, there is no statistical test on the agreement between human and automated evaluation.
>
> We apologize for the lack of clarity and not providing complete analysis results in the submitted version. We will incorporate the full details into the **main text**, and if space limitations arise, they will be included in the **Appendix**. Below, we provide the complete details and analysis results:
>
> We randomly sampled 50 instances for each scenario from the evaluation results of LLM(GPT-O4Mini), and validate the consistency between the evaluator and human evaluation using **three independent human annotators** with NLP expertise. The detailed annotation results can be found in the anonymous repository(https://anonymous.4open.science/r/ICF-Bench-B1C7).
>
> Description:
> - three scenarios: instructional forgetting(**IF**)、subtask revision(**SR**)、dynamic preference(**DP**)
> - Two task forms: **NoForget**、**Forget**
>
> Below are the average agreement rates (%) from three annotators:
>
> | Task |  NoForget | Forget |
> |------|----------------|--------------|
> | Instructional Forgetting | 89.33 | 93.33 |
> | Subtask Revision | 92.00 | 82.00 |
> | Dynamic Preference | 96.67 | 95.33 |
>
> Below are the respective agreement rates (%) from three annotators:
>
> | Task | NoForget(Annotator 1) | Forget(Annotator 1) | NoForget(Annotator 2) | Forget(Annotator 2) | NoForget(Annotator 3) | Forget(Annotator 3) |
> |:----:|:----------------------:|:-------------------:|:----------------------:|:-------------------:|:----------------------:|:-------------------:|
> | Instructional Forgetting | 88 | 92 | 90 | 94 | 90 | 94 |
> | Subtask Revision | 94 | 82 | 90 | 84 | 92 | 80 |
> | Dynamic Preference | 96 | 94 | 96 | 96 | 98 | 96 |
>
>
> To further strengthen the reliability of our human evaluation, we computed Cohen’s κ[1] for each annotator and Fleiss’ κ[2] across all three annotators.
>
> | Task | Cohen’s κ (Annotator 1) | Cohen’s κ (Annotator 2) | Cohen’s κ (Annotator 3) | Fleiss’ κ |
> |:------:|:------------------:|:------------------:|:------------------:|:-----------:|
> | IF_NoForget | 0.333 | 0.390 | 0.390 | 0.916 |
> | IF_Forget | 0.840 | 0.879 | 0.879 | 0.973 |
> | SR_NoForget | 0.841 | 0.749 | 0.891 | 0.870 |
> | SR_Forget | 0.633 | 0.672 | 0.711 | 0.918 |
> | DP_NoForget | 0.901 | 0.901 | 0.949 | 0.936 |
> | DP_Forget | 0.860 | 0.905 | 0.905 | 0.970 |
>
> - Overall, the agreement rates across the three annotators are consistently high (mostly above 90%，with slightly lower alignment in Subtask Revision above 80%), indicating clear evaluation criteria and low subjectivity.
> - Cohen’s κ for individual annotators ranges from 0.63 to 0.95, and Fleiss’ κ across all annotators remains above 0.87, reflecting a high level of consistency across annotators as well as between annotators and the automated evaluator(GPT-4OMini). The only exception is IF_NoForget, where Cohen’s κ is lower (0.33~0.39), which is likely caused by class imbalance, as the sampled items contain a high proportion of “True” labels.
> -  These results confirm that our automated evaluation framework is a valid and scalable proxy for human judgment.
>
>
> We will integrate these human evaluation results and analysis into **Section 4.5** and **Appendix** in the revised version.
>
> [1] Cohen J. A coefficient of agreement for nominal scales[J]. Educational and psychological measurement, 1960, 20(1): 37-46.
>
> [2] Fleiss J L. Measuring nominal scale agreement among many raters[J]. Psychological bulletin, 1971, 76(5): 378.
>
> ---
>
> >  Concerning readability, Table 1 and 2 are a bit hard to parse. Turning them into a figure (at least the average scores) could improve the readability of the paper. Also, combining the Tables might help see the trend across the metrics and avoid repeating the NA scores across the tables. The x labels in Figure 3 are not aligned.
>
> We appreciate the reviewer’s suggestion. In our revision paper, we will:
>
> - Merge Table 1 and Table 2 to avoid repetition.
> - Add a figure visualizing the average scores across models.
> - Fix the misaligned x-labels in Figure 3.
>
> Thanks for your advice, we think this will significantly improve the readability of our paper.
>
> ---
>
> >  Related work, you might find helpful the benchmark proposed in this paper [1] that is very similar to this work (subtask revision and dynamic preference scenarios).
>
> Thank you for pointing out this highly relevant work. Rakotonirina et al.[1] reveals that the current models have significant weaknesses in cross-session instruction memory and integration, which limits their application in real long-term collaboration. It feels very suitable for our work, and we will add it to our introduction section.
>
> [1] Rakotonirina, N. C., Hamdy, M., Campos, J. A., Weber, L., Testoni, A., Fadaee, M., Pezzele S. & Del Tredici, M. (2025). From Tools to Teammates: Evaluating LLMs in Multi-Session Coding Interactions. arXiv preprint arXiv:2502.13791.

---

> ### Author Response · Authors · 2025-11-21
> **Rebuttal (Page 3)**
>
> > Some times SFRR improves with model size (For example, SFRR goes from 33% for Llama3-8B to 46% for Llama3-70B), what causes the variations in SFRR across models?
>
> Thank you for the insightful comment. We agree that the relationship between model scale and SFRR is not monotonic and is influenced by multiple interacting factors. Although larger models typically achieve higher NoForget accuracy, this does not necessarily translate into higher SFRR. As shown in Table 2 in our paper, increasing model size sometimes improves average SFRR (e.g., Llama3-8B: 33.51% → Llama3-70B: 46.60%; Mistral-7B: 42,33% → Mistral-8x7B: 45.91%), but in other cases average SFRR decreases (e.g., Qwen2.5-7B: 42.09% → Qwen2.5-14B: 39.86%; Gemma3-27B(62.13%) exceeds both Llama3-70B(46.60%) and Qwen3-235B(50.80%)).
>
> We hypothesize that SFRR is influenced by a combination of different factors, rather than model size alone:
>
> - Training data distribution. Models trained on more dialogue data containing revisions, updates, contradictions, or preference shifts are likely to develop stronger in-context forgetting capabilities.
>
> - Instruction-following vs. consistency bias. Some large models exhibit stronger consistency or coherence preservation tendencies, which may make them less willing to override earlier context, even when the task requires forgetting.
>
> - Different alignment pipelines. Safety tuning and preference optimization (e.g., RLHF, DPO) may encourage either stricter adherence to earlier instructions or better adaptability to revised ones, leading to divergent SFRR outcomes across families.
>
> Overall, our findings suggest that model scale alone is not a reliable predictor of in-context forgetting performance, and SFRR is perhaps governed by data exposure, alignment strategies, among other factors.
>
> ---
>
> > Can the trends be described in a more quantitative way? what is the relation between the different metrics? scenarios?
>
> Thank you for the insightful suggestion. We have added a quantitative analysis to clarify the relationships between metrics and scenarios. Specifically, we computed Pearson correlations among NoForget Accuracy(**NA**), Forget Accuracy(**FA**), and Selective Forgetting Retention Rate (**SFRR**) in each scenario. The results are as follows:
>
> | Category | NA vs FA | NA vs SFRR | FA vs SFRR |
> |---------|---------------------|-------------------|------------------|
> | Instructional Forgetting | **0.504** | **0.536** | **0.999** |
> | Subtask Revision | **0.978** | **0.090** | **0.221** |
> | Dynamic Preference | **0.798** | **0.612** | **0.957** |
> | Average | **0.838** | **0.764** | **0.982** |
>
> These correlations reveal several key quantitative trends:
>
> - SFRR aligns more closely with Forget Accuracy than with NoForget Accuracy across all scenarios (average correlation 0.982 vs. 0.764). This supports  that SFRR primarily relates to robustness under interference rather than raw memory ability.
>
> - Instructional Forgetting scenario shows almost perfect FA–SFRR correlation (0.999), indicating that, under explicit “forget” instructions, SFRR is almost fully determined by interference sensitivity, not baseline memory.
>
> - Subtask Revision scenario exhibits a very different pattern, with extremely high NA–FA correlation (0.978) but almost no NA–SFRR correlation (0.090). This quantifies our observation that Revision tasks rely less on memorization accuracy and more on the model’s ability to integrate task updates, making SFRR behavior independent of baseline memory.
>
> - Dynamic Preference scenario yields high correlations across all pairs, with the highest correlation observed between FA and SFRR (0.957), followed by NoForget–Forget (0.798) and NoForget–SFRR (0.612), suggesting that preference following involves both  memory ability and interference sensitivity.
>
> We will incorporate these findings into the revised paper.
>
> > Could you provide more details about the human evaluation? To what extent is human evaluation in agreement with LLM evaluation?
>
> Thanks for your comment, this is fully addressed in weakness 2 above.
>
> ---
>
> We sincerely thank you again for your valuable feedback and hope our responses have addressed your concerns. To ensure the quality of the revised paper, we may need an additional 1–2 days after submitting this rebuttal to finalize the updates.
>
> Best regards,

---

> > ### Comment · Reviewer_XYBR · 2025-11-24
> >
> > Thank you for your detailed responses. Please make sure to include all the changes in the final version of the paper. As all my concerns were properly addressed, I have increased my score.

---

> > > ### Author Response · Authors · 2025-11-25
> > >
> > > Thank you very much for raising your score! We are glad that our clarifications addressed your concerns. We sincerely appreciate your thoughtful feedback and would like to assure you that we will incorporate all the requested changes. A revised version has already been submitted and is available for your review at your convenience.
> > >
> > > Best regards,

---

### Official Review · Reviewer_fQrB · 2025-10-23

**Soundness:** 3
**Presentation:** 3
**Contribution:** 2
**Rating:** 2
**Confidence:** 3

**Summary:**

This paper introduces "In-Context Forgetting" (ICF), defined as the ability of Large Language Models (LLMs) to selectively disregard interference information during inference, without any parameter updates. The paper's main contribution is ICF-Bench, the first comprehensive benchmark designed to evaluate this skill. The benchmark consists of 2,000 annotated, multi-turn dialogue samples covering three realistic scenarios: Instructional Forgetting, Subtask Revision, and Dynamic Preference. To measure performance, the authors use a paired-task format ("Noforget" vs. "Forget") and introduce a novel core metric: the Selective Forgetting Retention Rate (SFRR).

Key findings from testing various LLMs show:
1. All models perform well without interference but struggle significantly when interference is present.
2. Strong memory capacity does not guarantee strong ICF ability, revealing a fundamental "asymmetry" between the two skills.
3. The effect of context length on ICF varies by scenario.
These results expose critical vulnerabilities in current LLMs regarding adaptability, privacy, and user control.

**Strengths:**

1. The paper poses an interesting and previously underexplored problem. It defines in-context forgetting (ICF) clearly and distinguishes ICF from parameter-based machine unlearning.
2. The ICF-Bench is thoughtfully designed around three practical, real-world scenarios. The validation of the automated evaluator (GPT-04mini) against human agreement (80-98% correlation) confirms the reliability of the experimental results.
3. SFRR provides a fair measure of forgetting ability, as it isolates failures caused by interference from a model's general inability to perform the task. TThe paper's justification for SFRR (Appendix B) is convincing. The discovery of the "asymmetry" between memory and ICF is a valuable insight.

**Weaknesses:**

1. The term of "Forgetting" is debatable. As noted in the case study (Appendix C, lines 1026-1028), the model's response is "You asked me to forget... so I cannot recall them." There's more "Human forbidding sth." instead of "LLMs forgetting sth." This behavior is more likely to obeying an instruction to withhold information rather than a genuine "forgetting". The current evaluation may be measuring instruction compliance more than the intended forgetting mechanism.
2. The analysis of context length's impact is restricted to a single model (GPT-4O), limiting the generalizability of the findings. More importantly, as the authors themselves suggest in Section 4.4, poor performance on long-context tasks can sometimes mimic successful forgetting. It is difficult to classify whether a model 'forgets' due to following the instruction or due to its inherent limitations in long-context recall.
3. LLMs responses rated by LLMs could introduce potential bias. The paper would be strengthened by a brief discussion acknowledging this limitation and justifying the choice over other potential evaluation methods.
4. There're many typos in this paper.

**Questions:**

1. In Appendix B(line 916), the text refers to "Equation ??", which is a placeholder the authors forgot to fill in. It should likely refer to Equation (4).

2. Typos: "scenaios" is misspelled three times(line 224, line 249-250, line 367-368) and it should be "scenarios". "focuse on" in line 130 should be "focuses on". "The Figure demonstrate that" should be "The Figure demonstrates that" (line 249-250). "their ability to memory" shoule be "their ability to memorize" (line 094). "response persistently follow" should be "response persistently follows" (line 1083-1084). Moreover, "non negligible" would be better to be hyphenated as "non-negligible".

---

> ### Author Response · Authors · 2025-11-21
> **Rebuttal (Page 1)**
>
> Thank you for your time and effort in reviewing our work. We would like to address the questions you raised as follows:
>
> ---
> > The term of "Forgetting" is debatable. As noted in the case study (Appendix C, lines 1026-1028), the model's response is "You asked me to forget... so I cannot recall them." There's more "Human forbidding sth." instead of "LLMs forgetting sth." This behavior is more likely to obeying an instruction to withhold information rather than a genuine "forgetting". The current evaluation may be measuring instruction compliance more than the intended forgetting mechanism.
>
> Thank you for the comment. Please let me address your concern as follows:
>
> - **"Forgetting" in our paper does not imply permanent erasure of information.** We agree that in-context forgetting inherently involves **instruction following**. As mentioned in Reviewer XYBR’s summary, *"The paper proposes ICF-Bench, a benchmark that measures Large Language Models (LLMs)' ability to follow instructions while taking into account explicit or implicit updates, called forget interference, in the conversation."* This is precisely the capability our benchmark evaluates. **In-context forgetting does not require LLMs to truly forget some information** (which would require parameter updates).  Truly forgetting information is related to **machine unlearning**,  which we have already discussed in the related work section (Machine Unlearning subsection), where **we clearly explained how it differs from our work**.
> - **"Forgetting" in our paper is mainly reflected in the realistic scenarios we are involved in**. As mentioned in Reviewer bYng’s summary,  *"Many prior works study and prioritise memory retention and long-context reasoning, however, this paper argues that real interactions also require models to discard outdated or conflicting context, with implications for privacy, personalization, and robustness."*. Although existing work has extensively examined the memorization and utilization of prior context, **much less is known about whether LLMs can effectively forget outdated, inconsistent with preference, or explicitly discarded information which is often appears in realistic interactions**, and **subsequently generate responses that align with user's expectations**. This gap is precisely what ICF-Bench aims to address.
> - Finally, what we want to clarify is the example response *"You asked me to forget... so I cannot recall them."* raised in your comment. When we conduct the evaluation, we consider it as **correct forgetting because it meets the user's expectation, forgetting or not answering the discarded information**.
>
> We will further clarify this distinction in our revised paper to avoid potential ambiguity.
>
> ---
>
> >  The analysis of context length's impact is restricted to a single model (GPT-4O), limiting the generalizability of the findings.
>
> This is a very reasonable suggestion. To address this, we extended our context-length experiments to include:
>
> - **Mixtral8x7B** (32k context),
> - **LLAMA3.3-70B** (The original llama3-70B only supported 8k, so it was changed to llama3.3-70B which supports 128k),
> - original **GPT-4o** results (**the same as the submitted paper**).
> - scenarios: instructional forgetting(**IF**)、subtask revision(**SR**)、dynamic preference(**DP**)
> - metrics: NoForget Accuracy(**NA**)、**Forget Accuracy**(**FA**)、Selective Forgetting Retention Rate(**SFRR**)
>
> **Mixtral-8x7B**
>
> | Context | IF_NA | IF_FA | IF_SFRR | SR_NA | SR_FA | SR_SFRR | DP_NA | DP_FA | DP_SFRR |
> |---------|-------------|-----------|---------|-------------|-----------|---------|-------------|-----------|---------|
> | 500 | 87.07 | 15.13 | 16.68 | 63.42 | 52.84 | 74.30 | 59.67 | 36.44 | 52.32 |
> | 1000 | 86.23 | 14.45 | 15.61 | 61.53 | 51.12 | 73.29 | 58.43 | 32.10 | 50.18 |
> | 3000 | 84.12 | 11.10 | 12.58 | 58.28 | 49.40 | 74.73 | 54.18 | 28.58 | 47.20 |
> | 6000 | 80.52 | 9.16 | 10.54 | 54.33 | 46.58 | 75.28 | 50.88 | 23.12 | 44.03 |
> | 10000 | 77.30 | 7.96 | 9.10 | 49.63 | 43.12 | 76.63 | 47.12 | 18.10 | 36.18 |
> | 15000 | 73.42 | 7.02 | 8.03 | 44.48 | 40.53 | 78.18 | 42.53 | 13.82 | 30.48 |
> | 30000 | 69.78 | 5.31 | 6.29 | 39.13 | 37.63 | 80.38 | 36.92 | 10.28 | 24.42 |
>
> **Llama3.3-70B**
> | Context | IF_NA | IF_FA | IF_SFRR | SR_NA | SR_FA | SR_SFRR | DP_NA | DP_FA | DP_SFRR |
> |---------|-------------|-----------|---------|-------------|-----------|---------|-------------|-----------|---------|
> | 500 | 94.47 | 15.19 | 15.43 | 61.71 | 51.37 | 79.97 | 75.34 | 40.37 | 49.93 |
> | 1000 | 93.71 | 14.24 | 14.12 | 60.13 | 49.87 | 80.62 | 73.88 | 36.42 | 46.50 |
> | 3000 | 91.65 | 12.59 | 13.02 | 56.37 | 47.96 | 82.22 | 69.41 | 32.83 | 45.70 |
> | 6000 | 88.30 | 11.40 | 11.96 | 52.93 | 45.45 | 83.02 | 65.69 | 29.15 | 43.26 |
> | 10000 | 85.11 | 10.88 | 11.21 | 48.61 | 42.63 | 84.19 | 61.17 | 20.16 | 33.84 |
> | 15000 | 81.32 | 9.56 | 9.78 | 43.08 | 40.13 | 85.48 | 55.87 | 17.22 | 31.14 |
> | 30000 | 77.82 | 7.39 | 7.75 | 37.95 | 37.48 | 87.26 | 48.64 | 12.27 | 25.12 |

---

> ### Author Response · Authors · 2025-11-21
> **Rebuttal (Page 2)**
>
> **Analysis across models:** The context‑length ablation shows **consistent trends**:
> - **NoForget Accuracy** and **Forget Accuracy** show a decreasing trend as the context length increases in three scenarios.
> - **Instructional forgetting** and **dynamic preference** tend to **suffer** (lower IF_SFRR and DP_SFRR) as the context length increases (attention dilution and unstable recency effects).
> - **Subtask revision** tends to **benefit** (higher SR_SFRR) as the context length increases (reduced influence of earlier subtask signals).
>
> We will integrate this cross‑model results and analysis into **Section 4.4** and **Appendix** in the revised version.
>
> ---
>
> > More importantly, as the authors themselves suggest in Section 4.4, poor performance on long-context tasks can sometimes mimic successful forgetting. It is difficult to classify whether a model 'forgets' due to following the instruction or due to its inherent limitations in long-context recall.
>
> Thank you for the comment. Please let me address your concern as follows:
> - **First, we would like to clarify that our intention was not to suggest that *"poor performance on long-context tasks can mimic successful forgetting"* in Section 4.4.** This chapter mainly discusses the influence of context length on the ability of in-context forgetting and the different trend of the SFRR metric with context length in the subtask revision scenario. We hypothesise that as the context becomes longer, the influence of early information on the revision task diminishes(**recency bias**[1]), which leads to an upward trend in SFRR. **It is not suggested that memory defects would be regarded as successful forgetting**.
>
> - **Second, we are also deeply concerned about the issue that the memory defect of the model itself may be regarded as successful forgetting**. In the discussion of "Why not rely solely on Forget Accuracy?"(Appendix B), we report **the proportion of cases where models answered incorrectly without forgetting interference but correctly with forgetting interference**. This highlights a key limitation of using raw Forget Accuracy as the sole metric for evaluating in-context forgetting.
>
> - **Finally, we propose the Selective Forgetting Retention Rate (SFRR) to resolve this issue.** As mentioned by your strength 3, *"SFRR provides a fair measure of forgetting ability, as it isolates failures caused by interference from a model's general inability to perform the task. The paper's justification for SFRR (Appendix B) is convincing. The discovery of the "asymmetry" between memory and ICF is a valuable insight."*, which **measures the proportion of originally correct responses that remain correct in the presence of forgetting interference**. This ensures that only instances in which the model truly retains the necessary information while still needing to forget interference are counted. **One of the primary purposes of proposing SFRR is to avoid cases where a model’s memory defects is incorrectly regarded as successful in-context forgetting.**
>
> We will revise Section 4.4(e.g., by removing the last sentence, *"as the model’s limited robustness to long contexts sometimes leads to improved performance in ICF"*) to avoid potential ambiguity and highlight the role of the Selective Forgetting Retention Rate (SFRR) to resolve the issue that memory defects would be regarded as successful forgetting.
>
> [1] Clark C, Oh B D, Schuler W. Linear recency bias during training improves transformers’ fit to reading times[C]//Proceedings of the 31st international conference on computational linguistics. 2025: 7735-7747.
>
> ---
>
> >  LLMs responses rated by LLMs could introduce potential bias. The paper would be strengthened by a brief discussion acknowledging this limitation and justifying the choice over other potential evaluation methods.
>
> Thank you for pointing this out. Although using LLMs for evaluation is a common practice in both academia and industry, we will acknowledge that this approach may introduce potential bias in **Section 4.5** and explain the rationale behind our design:
>
> 1. **Generative answers lack deterministic gold labels**, making classical quantitative metrics(e.g., EM、F1、Rouge) inappropriate. We also try to improve the prompt used for evaluation(**Appendix A**).
> 2. We avoid using any **evaluated** model as the **evaluator** (e.g., GPT-4o, GPT-5, GPT-O3Mini), so we adopt **GPT-O4mini**, which is inexpensive, fast, and sufficiently capable.
> 3. We validate the consistency between the evaluator and human evaluation using **three independent human annotators** with NLP expertise. The original annotation results can be found in the anonymous repository (https://anonymous.4open.science/r/ICF-Bench-B1C7) and the complete details and analysis results as follows:

---

> ### Author Response · Authors · 2025-11-21
> **Rebuttal (Page 3)**
>
> Description:
> - three scenarios: instructional forgetting(**IF**)、subtask revision(**SR**)、dynamic preference(**DP**)
> - Two task forms: **NoForget**、**Forget**
>
> Below are the average agreement rates (%) from three annotators:
>
> | Task |  NoForget | Forget |
> |------|----------------|--------------|
> | Instructional Forgetting | 89.33 | 93.33 |
> | Subtask Revision | 92.00 | 82.00 |
> | Dynamic Preference | 96.67 | 95.33 |
>
> Below are the respective agreement rates (%) from three annotators:
>
> | Task | NoForget(Annotator 1) | Forget(Annotator 1) | NoForget(Annotator 2) | Forget(Annotator 2) | NoForget(Annotator 3) | Forget(Annotator 3) |
> |:----:|:----------------------:|:-------------------:|:----------------------:|:-------------------:|:----------------------:|:-------------------:|
> | Instructional Forgetting | 88 | 92 | 90 | 94 | 90 | 94 |
> | Subtask Revision | 94 | 82 | 90 | 84 | 92 | 80 |
> | Dynamic Preference | 96 | 94 | 96 | 96 | 98 | 96 |
>
>
>
> To further strengthen the reliability of our human evaluation, we computed Cohen’s κ[1] for each annotator and Fleiss’ κ[2] across all three annotators.
>
> | Task | Cohen’s κ (Annotator 1) | Cohen’s κ (Annotator 2) | Cohen’s κ (Annotator 3) | Fleiss’ κ |
> |:------:|:------------------:|:------------------:|:------------------:|:-----------:|
> | IF_NoForget | 0.333 | 0.390 | 0.390 | 0.916 |
> | IF_Forget | 0.840 | 0.879 | 0.879 | 0.973 |
> | SR_NoForget | 0.841 | 0.749 | 0.891 | 0.870 |
> | SR_Forget | 0.633 | 0.672 | 0.711 | 0.918 |
> | DP_NoForget | 0.901 | 0.901 | 0.949 | 0.936 |
> | DP_Forget | 0.860 | 0.905 | 0.905 | 0.970 |
>
> - Overall, the agreement rates across the three annotators are consistently high (mostly above 90%，with slightly lower alignment in Subtask Revision above 80%), indicating clear evaluation criteria and low subjectivity.
> - Cohen’s κ for individual annotators ranges from 0.63 to 0.95, and Fleiss’ κ across all annotators remains above 0.87, reflecting a high level of consistency across annotators as well as between annotators and the automated evaluator(GPT-4OMini). The only exception is IF_NoForget, where Cohen’s κ is lower (0.33~0.39), which is likely caused by class imbalance, as the sampled items contain a high proportion of “True” labels.
> -  These results confirm that our automated evaluation framework is a valid and scalable proxy for human judgment.
>
> We will integrate these human evaluation results and analysis into **Section 4.5** and **Appendix** in the revised version.
>
> [1] Cohen J. A coefficient of agreement for nominal scales[J]. Educational and psychological measurement, 1960, 20(1): 37-46.
>
> [2] Fleiss J L. Measuring nominal scale agreement among many raters[J]. Psychological bulletin, 1971, 76(5): 378.
>
> ---
>
> > There're many typos in this paper.
>
> We sincerely apologize for the oversights. All issues identified by your comment have been corrected, including:
> - *scenaios → scenarios*
> - *focuse on → focus on*
> - *The Figure demonstrate → demonstrates*
> - *their ability to memory → memorize*
> - *response persistently follow → follows*
> - *non negligible → non-negligible*
> - missing reference in Appendix B (*Equation ?? → Equation (4)*)
>
> We also will conduct a thorough, full-manuscript review of the revised version to prevent such issues from occurring again.
>
> ---
> We thank you again for your valuable feedback and would greatly appreciate it if you could reconsider your evaluation should our responses address your concerns. To ensure the quality of the revised paper(we apologize again for the oversights about typos), we may need an additional 1–2 days after submitting this rebuttal to finalize the updates.
>
> Best regards,

---

> > ### Comment · Reviewer_fQrB · 2025-11-24
> >
> > Thank you for your response, it is helpful and I have raised my score from 2->4.

---

> > > ### Author Response · Authors · 2025-11-25
> > >
> > > Thank you very much for raising your score! We are glad that our clarifications addressed your concerns. If you have any further questions or if anything remains unclear, please let us know and we will be happy to provide additional explanation.
> > >
> > > Best regards,

---

### Official Review · Reviewer_bYng · 2025-10-31

**Soundness:** 3
**Presentation:** 3
**Contribution:** 3
**Rating:** 6
**Confidence:** 4

**Summary:**

This paper studies whether large language models can selectively forget information during inference, a capability the authors call in-context forgetting (ICF). Many prior works study and prioritise memory retention and long-context reasoning, however, this paper argues that real interactions also require models to discard outdated or conflicting context, with implications for privacy, personalization, and robustness. The authors introduce ICF-Bench, a benchmark of 2,000 multi-turn dialogues, and each example has a No-Forget and Forget condition. The main metric, Selective Forgetting Retention Rate (SFRR), measures how often models that answered correctly without interference remain correct when forgetting is required.

Experiments across recent proprietary and open models show that models perform well without interference but degrade substantially when forgetting is required. The experiments find that a higher memory retention does not imply strong forgetting ability, revealing an asymmetry between remembering and forgetting.

**Strengths:**

- The paper identifies a meaningful gap: models can remember well but struggle to intentionally forget, which matters for privacy, preference updates, and real conversational robustness. Framing in-context forgetting as a capability pushes beyond typical long-context evaluation.

- Well designed benchmark with three representative scenarios of ICF: Instructional Forgetting, Subtask Revision, Dynamic Preference, with Forget and NoForget alternatives. This is a clean evaluation structure that avoids confounding memory quality with forgetting ability.

- The demonstrated dissociation between memory strength and forgetting ability is useful. It gives empirical proof to a hypothesis most practitioners may have suspected but not quantified.

**Weaknesses:**

- The paper varies context length within conversational tasks (by prepending other conversations), yet context dynamics in dialogue are only one class of real-world usage. Long-form reasoning, tool-augmented workflows, and multi-document tasks involve qualitatively different context structures and memory demands. As a result, the observed context-length effects may not generalize beyond dialogue settings. Expanding or at least discussing evaluations in these alternative settings would strengthen claims of generality.

**Questions:**

1) Several related work are mentioned in subsection "In-context Unlearning and Context Management." - what are the main limitations in the evaluation methods used in these papers? A more complete discussion of existing evaluation techniques in the literature would strengthen the paper and it's positioning.

---

> ### Author Response · Authors · 2025-11-21
> **Rebuttal (Page 1)**
>
> Thank you for your time and effort in reviewing our work. We would like to address the questions you raised as follows:
>
> ---
>
> > The paper varies context length within conversational tasks (by prepending other conversations), yet context dynamics in dialogue are only one class of real-world usage. Long-form reasoning, tool-augmented workflows, and multi-document tasks involve qualitatively different context structures and memory demands. As a result, the observed context-length effects may not generalize beyond dialogue settings. Expanding or at least discussing evaluations in these alternative settings would strengthen claims of generality.
>
> We appreciate this insightful observation and fully acknowledge that our current experiments on context length variation are only conducted within multi-turn dialogue environments, representing a highly dynamic type of long-context usage.
>
> This design choice is intentional: dialogue naturally involves frequent **context overwriting, preference changes, and semantic entanglement** between turns, making it a particularly sensitive and realistic setting for evaluating in-context forgetting performance.
>
> While our present findings are based on dialogue, the trends observed across different context lengths (further supported by additional experiments on Mixtral-8×7B (32k) and Llama3.3-70B (128k) beyond GPT-4o. Supplementary results can be found at the end of this response.) reflect underlying phenomena such as **attention dilution**[1] and **recency bias**[2], which are architectural rather than task-specific. These mechanisms are also relevant to long-form reasoning and multi-document understanding.
>
> In future work, we plan to extend ICF-Bench beyond dialogue, including tasks that require sustained reasoning across documents and tool-augmented pipelines. We will **clarify this limitation that ICF-Bench currently focuses solely on in-dialogue evaluation** and our expansion plans in the revised paper, to better delineate the scope of our current findings.
>
> To further substantiate our findings regarding the impact of in-dialogue context length on in-context forgetting performance, we extended our context-length  experiments to include:
>
> - **Mixtral8x7B** (32k context),
> - **LLAMA3.3-70B** (The original llama3-70B only supported 8k, so it was changed to llama3.3-70B which supports 128k),
> - the original **GPT-4o** results (**the same as the submitted paper**).
> - three scenarios: instructional forgetting(**IF**)、subtask revision(**SR**)、dynamic preference(**DP**)
> - three metrics: NoForget Accuracy(**NA**)、**Forget Accuracy**(**FA**)、Selective Forgetting Retention Rate(**SFRR**)
>
> **Mixtral-8x7B**
>
> | Context | IF_NA | IF_FA | IF_SFRR | SR_NA | SR_FA | SR_SFRR | DP_NA | DP_FA | DP_SFRR |
> |---------|-------------|-----------|---------|-------------|-----------|---------|-------------|-----------|---------|
> | 500 | 87.07 | 15.13 | 16.68 | 63.42 | 52.84 | 74.30 | 59.67 | 36.44 | 52.32 |
> | 1000 | 86.23 | 14.45 | 15.61 | 61.53 | 51.12 | 73.29 | 58.43 | 32.10 | 50.18 |
> | 3000 | 84.12 | 11.10 | 12.58 | 58.28 | 49.40 | 74.73 | 54.18 | 28.58 | 47.20 |
> | 6000 | 80.52 | 9.16 | 10.54 | 54.33 | 46.58 | 75.28 | 50.88 | 23.12 | 44.03 |
> | 10000 | 77.30 | 7.96 | 9.10 | 49.63 | 43.12 | 76.63 | 47.12 | 18.10 | 36.18 |
> | 15000 | 73.42 | 7.02 | 8.03 | 44.48 | 40.53 | 78.18 | 42.53 | 13.82 | 30.48 |
> | 30000 | 69.78 | 5.31 | 6.29 | 39.13 | 37.63 | 80.38 | 36.92 | 10.28 | 24.42 |
>
>
> **Llama3.3-70B**
> | Context | IF_NA | IF_FA | IF_SFRR | SR_NA | SR_FA | SR_SFRR | DP_NA | DP_FA | DP_SFRR |
> |---------|-------------|-----------|---------|-------------|-----------|---------|-------------|-----------|---------|
> | 500 | 94.47 | 15.19 | 15.43 | 61.71 | 51.37 | 79.97 | 75.34 | 40.37 | 49.93 |
> | 1000 | 93.71 | 14.24 | 14.12 | 60.13 | 49.87 | 80.62 | 73.88 | 36.42 | 46.50 |
> | 3000 | 91.65 | 12.59 | 13.02 | 56.37 | 47.96 | 82.22 | 69.41 | 32.83 | 45.70 |
> | 6000 | 88.30 | 11.40 | 11.96 | 52.93 | 45.45 | 83.02 | 65.69 | 29.15 | 43.26 |
> | 10000 | 85.11 | 10.88 | 11.21 | 48.61 | 42.63 | 84.19 | 61.17 | 20.16 | 33.84 |
> | 15000 | 81.32 | 9.56 | 9.78 | 43.08 | 40.13 | 85.48 | 55.87 | 17.22 | 31.14 |
> | 30000 | 77.82 | 7.39 | 7.75 | 37.95 | 37.48 | 87.26 | 48.64 | 12.27 | 25.12 |
>
> **Analysis across models:** The context‑length ablation shows **consistent trends**:
> - **NoForget Accuracy** and **Forget Accuracy** show a decreasing trend as the context length increases in three scenarios.
> - **Instructional forgetting** and **dynamic preference** tend to **suffer** (lower IF_SFRR and DP_SFRR) as the context length increases (attention dilution and unstable recency effects).
> - **Subtask revision** tends to **benefit** (higher SR_SFRR) as the context length increases (reduced influence of earlier subtask signals).

---

> ### Author Response · Authors · 2025-11-21
> **Rebuttal (Page 2)**
>
> [1] Zhang X, Chang X, Li M, et al. Selective attention: Enhancing transformer through principled context control[J]. Advances in Neural Information Processing Systems, 2024, 37: 11061-11086.
>
> [2] Clark C, Oh B D, Schuler W. Linear recency bias during training improves transformers’ fit to reading times[C]//Proceedings of the 31st international conference on computational linguistics. 2025: 7735-7747.
>
> ---
>
> > Several related work are mentioned in subsection "In-context Unlearning and Context Management." - what are the main limitations in the evaluation methods used in these papers? A more complete discussion of existing evaluation techniques in the literature would strengthen the paper and it's positioning.
>
> Thank you for highlighting the need for a deeper discussion of related evaluation methods. These works cited in our "In-context Unlearning and Context Management" section aim to demonstrate that **some researchers are attempting to enable models to selectively forget or compress interfering information (e.g.,sensitive, outdated, or irrelevant information) during inference**, supporting the significance of ICF-Bench as the comprehensive benchmark for evaluating in-context forgetting in real dialogue scenarios.
>
> The evaluation methodologies used in these works differ substantially from ours:
> - Pawelczyk et al. [1] and Zhang et al. [2] follow evaluation protocols similar to those in machine unlearning by constructing a retain set and a forget set, and then measuring response accuracy on the retain set and refusal rates on the forget set. Such approaches are well suited for assessing whether a model’s memory of specific training samples has been “removed” or “suppressed.” **However, their evaluations primarily focus on training-sample-level forgetting, rather than on forgetting dynamic interference information within a dialogue.**
> - Lin et al. [3] compare standard Transformers across downstream short-context tasks (e.g., WikiText, LAMBADA, QA tasks) and long-context tasks (LongBench), and analyze the model’s ability to reuse key information through the “needle-in-a-haystack” test.
> - Ge et al. [4], Dai et al. [5], and Wang et al. [6] evaluate models by examining the degree of context compression during inference and the resulting downstream task accuracy.
>
> These works primarily evaluate the effectiveness of **forgetting static information** or **compressing context without forgetting**, rather than a model’s ability to **dynamically manage information** (retain or forget) based on the evolving context. In contrast, our proposed ICF-Bench is specifically designed to evaluate an LLM’s **dynamic forgetting capability** in realistic dialogue settings, where the context may contain sensitive, outdated, or irrelevant interfering information.
>
> Thanks for your comment, we will expand the discussion in the "In-context Unlearning and Context Management" subsection to **highlight these difference and clarify our contribution** in our revised paper.
>
> [1]Pawelczyk M, Neel S, Lakkaraju H. In-context unlearning: Language models as few shot unlearners[J]. arXiv preprint arXiv:2310.07579, 2023.
>
> [2]Takashiro S, Kojima T, Gambardella A, et al. Answer when needed, forget when not: Language models pretend to forget via in-context knowledge unlearning[C]//Findings of the Association for Computational Linguistics: ACL 2025. 2025: 24872-24885.
>
> [3]Lin Z, Nikishin E, He X O, et al. Forgetting transformer: Softmax attention with a forget gate[J]. arXiv preprint arXiv:2503.02130, 2025.
>
> [4]Ge T, Hu J, Wang L, et al. In-context autoencoder for context compression in a large language model[J]. arXiv preprint arXiv:2307.06945, 2023.
>
> [5]Dai Y, Lian J, Huang Y, et al. Pretraining context compressor for large language models with embedding-based memory[C]//Proceedings of the 63rd Annual Meeting of the Association for Computational Linguistics (Volume 1: Long Papers). 2025: 28715-28732.
>
> [6]Wang X, Chen Z, Xu T, et al. In-context former: Lightning-fast compressing context for large language model[C]//Findings of the Association for Computational Linguistics: EMNLP 2024. 2024: 2445-2460.
>
> ---
> We sincerely thank you again for your valuable feedback and hope our responses have addressed your concerns. To ensure the quality of the revised paper, we may need an additional 1–2 days after submitting this rebuttal to finalize the updates.
>
> Best regards,

---

### Official Review · Reviewer_nisY · 2025-11-02

**Soundness:** 3
**Presentation:** 3
**Contribution:** 3
**Rating:** 8
**Confidence:** 4

**Summary:**

This paper introduces ICF-Bench, a benchmark for evaluating in-context forgetting (ICF) in large language models, which is the ability to discard outdated or contradicted information within a conversation. Unlike prior work focusing on memorization or unlearning, the authors design controlled multi-turn dialogues where models must selectively “forget” previous content when given new or conflicting instructions. They evaluate several open and proprietary LLMs across three scenarios (instructional forgetting, subtask revision, dynamic preference) using paired tasks with and without interference. Results show that current LLMs perform well without interference but degrade sharply when forgetting is required, illustrating vulnerabilities of current LLMs in terms of privacy protection, adaptability, and user autonomy.

**Strengths:**

1. The paper focuses on an important dimension of LLM behavior, the ability to forget selectively, which is particularly relevant for user privacy, adaptability, and evolving dialogue contexts.
2. The benchmark design is systematic and well-structured, covering three realistic dialogue scenarios with paired no-interference vs interference tasks.
3. The experimental evaluation is comprehensive, including multiple open-source and proprietary models, context-length variation, and a human evaluation for the automated metric. This offers convincing evidence of the claimed vulnerabilities.

**Weaknesses:**

1. The instructional forgetting tasks might be narrowly defined. While it provides a reasoning measurement of LLMs to follow precise forgetting instructions, it's unclear the value of optimizing this specific capability for LLMs.
2. The paper discusses “forgetting” mainly as suppression of prior information, but less on why the model fails (e.g., attention routing, representation entanglement) and on concrete architectural remedies.
3. The ablation on the impact of context length is rather shallow. The experiment is only conducted for GPT-4o. It's unclear whether the conclusion drew here would generalize.

**Questions:**

1. Have the authors experimented prompt engineering to improve ICF ability even preliminarily?

---

> ### Author Response · Authors · 2025-11-21
> **Rebuttal (Page 1)**
>
> Thank you for your time and effort in reviewing our work. We would like to address the questions you raised as follows:
>
> ---
>
> >  1. The instructional forgetting tasks might be narrowly defined. While it provides a reasoning measurement of LLMs to follow precise forgetting instructions, it's unclear the value of optimizing this specific capability for LLMs.
>
> Thank you for the comment. We agree the paper should emphasize the practical value of improving the instructional forgetting task. In real applications, users frequently need LLMs to explictly or implictly “forget” **sensitive or outdated information mentioned in historical dialogues**(e.g., personal details, incorrect assumptions, or information that do not wish to be referenced in the current turn). Failing to comply with such instructions may lead to privacy exposure or task inconsistency.
>
> Thus, we consider optimizing the instructional forgetting task to be a crucial step for **enabling users to actively control LLMs in managing context**, and we include it as a core scenario in our benchmark. This capability is closely related to **personalization**, **privacy protection** and **user autonomy**.
>
> We will add an explanation of this motivation in our revised paper to better highlight its practical importance.
>
> ---
>
> > 2. The paper discusses “forgetting” mainly as suppression of prior information, but less on why the model fails (e.g., attention routing, representation entanglement) and on concrete architectural remedies.
>
> Thank you for this insightful comment. We fully agree that understanding the underlying reasons for ICF performance issues and exploring architectural improvements is a critical next step.
>
> Our contribution in the current paper is a comprehensive benchmark (ICF-Bench) and controlled paired evaluations to reveal capability gaps between memory and ICF rather than analyzing internal mechanisms.
>
> Nevertheless, we have begun to analyze the possible reasons behind the observed limitations in our paper:
>
> - "Instead, these findings suggest that in-context forgetting requires distinct cognitive mechanisms, potentially involving dynamic attention reallocation, conflict resolution between competing contextual signals, and robust suppression of outdated information, which are not automatically enhanced through scaling alone."(Chapter 4.3, Paragraph 1)
> - "This indicates a qualitative improvement in tracking and prioritizing evolving user intents, likely due to enhanced preference modeling in training."(Chapter 4.3, Paragraph 2)
> - "Overall, although long contexts generally exacerbate the challenges of in-context forgetting, they can mitigate the influence of early context in revision tasks, as the model’s limited robustness to long contexts sometimes leads to improved performance in ICF."(Chapter 5)
>
> Based on your suggestion, we will add some discussion on the underlying causes of model performance and expand Sections 4.2 and 4.4 to analyze these phenomena and their root mechanisms. For example:
> - After introducing forgetting interference, models generally perform worse compared to without forgetting interference. This may be because standard attention training objectives inherently prioritize retaining and leveraging historical information (favoring recall or continuation), and thus lack explicit “withdraw or delete” pathways. As a result, attention reallocation under conflict exhibits **attention inertia**[1], and different types of information (facts, subtasks, preferences) overlap in the vector space. This **representation entanglement**[2] makes simple natural language negation insufficient for the model to distinguish and discard specific components during retrieval, leading to a general inability to suppress or revoke prior content.
> - As context length increases, different scenarios exhibit divergent trends, likely due to the combined effects of **attention dilution**[3] and **recency bias**[4]. In dynamic preference, attention dilution weakens the relative strength of the latest preference, while recency bias becomes unstable with increasing distance, making the model more likely to revert to earlier preferences and causing performance degradation. In subtask revision, however, attention decay caused by distance reduces the influence of old subtasks, making it easier for the model to adopt the new instruction, thereby improving SFRR. The trend of instructional forgetting is similar to that of dynamic preference. These mechanisms collectively explain the directional differences in forgetting performance under long-context conditions.
>
> In the revised version, we will propose exploring **how to enhance LLMs' in-context forgetting performance as future work**, including prompt engineering, attention-level modulation techniques, representation disentanglement, architectural forgetting gates，among other factors.

---

> ### Author Response · Authors · 2025-11-21
> **Rebuttal (Page 2)**
>
> [1] Hosseinzadeh R, Sadeghzadeh M. Attention Mechanisms in Transformers: A General Survey[J]. Journal of AI and Data Mining, 2025, 13(3): 359-368.
>
> [2] Wu G, Zhang S, Shi R, et al. Representation Entanglement for Generation: Training Diffusion Transformers Is Much Easier Than You Think[J]. arXiv preprint arXiv:2507.01467, 2025.
>
> [3] Zhang X, Chang X, Li M, et al. Selective attention: Enhancing transformer through principled context control[J]. Advances in Neural Information Processing Systems, 2024, 37: 11061-11086.
>
> [4] Clark C, Oh B D, Schuler W. Linear recency bias during training improves transformers’ fit to reading times[C]//Proceedings of the 31st international conference on computational linguistics. 2025: 7735-7747.
>
> ---
>
> > 3. The ablation on the impact of context length is rather shallow. The experiment is only conducted for GPT-4o. It's unclear whether the conclusion drew here would generalize.
>
> Thank you for this very reasonable comment. To address this, we extended our context-length experiments to include:
>
> - **Mixtral8x7B** (32k context),
> - **LLAMA3.3-70B** (The original llama3-70B only supported 8k, so it was changed to llama3.3-70B which supports 128k),
> - the original **GPT-4o** results.
> - three scenarios: instructional forgetting(**IF**)、subtask revision(**SR**)、dynamic preference(**DP**)
> - three metrics: NoForget Accuracy(**NA**)、**Forget Accuracy**(**FA**)、Selective Forgetting Retention Rate(**SFRR**)
>
> **Mixtral-8x7B**
>
> | Context | IF_NA | IF_FA | IF_SFRR | SR_NA | SR_FA | SR_SFRR | DP_NA | DP_FA | DP_SFRR |
> |---------|-------------|-----------|---------|-------------|-----------|---------|-------------|-----------|---------|
> | 500 | 87.07 | 15.13 | 16.68 | 63.42 | 52.84 | 74.30 | 59.67 | 36.44 | 52.32 |
> | 1000 | 86.23 | 14.45 | 15.61 | 61.53 | 51.12 | 73.29 | 58.43 | 32.10 | 50.18 |
> | 3000 | 84.12 | 11.10 | 12.58 | 58.28 | 49.40 | 74.73 | 54.18 | 28.58 | 47.20 |
> | 6000 | 80.52 | 9.16 | 10.54 | 54.33 | 46.58 | 75.28 | 50.88 | 23.12 | 44.03 |
> | 10000 | 77.30 | 7.96 | 9.10 | 49.63 | 43.12 | 76.63 | 47.12 | 18.10 | 36.18 |
> | 15000 | 73.42 | 7.02 | 8.03 | 44.48 | 40.53 | 78.18 | 42.53 | 13.82 | 30.48 |
> | 30000 | 69.78 | 5.31 | 6.29 | 39.13 | 37.63 | 80.38 | 36.92 | 10.28 | 24.42 |
>
>
> **Llama3.3-70B**
> | Context | IF_NA | IF_FA | IF_SFRR | SR_NA | SR_FA | SR_SFRR | DP_NA | DP_FA | DP_SFRR |
> |---------|-------------|-----------|---------|-------------|-----------|---------|-------------|-----------|---------|
> | 500 | 94.47 | 15.19 | 15.43 | 61.71 | 51.37 | 79.97 | 75.34 | 40.37 | 49.93 |
> | 1000 | 93.71 | 14.24 | 14.12 | 60.13 | 49.87 | 80.62 | 73.88 | 36.42 | 46.50 |
> | 3000 | 91.65 | 12.59 | 13.02 | 56.37 | 47.96 | 82.22 | 69.41 | 32.83 | 45.70 |
> | 6000 | 88.30 | 11.40 | 11.96 | 52.93 | 45.45 | 83.02 | 65.69 | 29.15 | 43.26 |
> | 10000 | 85.11 | 10.88 | 11.21 | 48.61 | 42.63 | 84.19 | 61.17 | 20.16 | 33.84 |
> | 15000 | 81.32 | 9.56 | 9.78 | 43.08 | 40.13 | 85.48 | 55.87 | 17.22 | 31.14 |
> | 30000 | 77.82 | 7.39 | 7.75 | 37.95 | 37.48 | 87.26 | 48.64 | 12.27 | 25.12 |
>
>
> **GPT-4o**
> | Context | IF_NA | IF_FA | IF_SFRR | SR_NA | SR_FA | SR_SFRR | DP_NA | DP_FA | DP_SFRR |
> |---------|-------------|-----------|---------|-------------|-----------|---------|-------------|-----------|---------|
> | 500     | 96.53       | 59.43     | 59.09   | 83.91       | 65.53     | 71.76   | 90.36       | 52.83     | 52.46   |
> | 1000    | 95.53       | 57.24     | 55.82   | 81.16       | 62.92     | 70.29   | 89.35       | 46.33     | 50.85   |
> | 3000    | 93.53       | 50.60     | 51.61   | 77.49       | 60.61     | 72.88   | 81.44       | 43.17     | 51.08   |
> | 6000    | 89.05       | 46.21     | 48.07   | 72.90       | 57.11     | 73.61   | 76.69       | 38.71     | 48.47   |
> | 10000   | 85.47       | 44.64     | 46.13   | 67.09       | 53.03     | 75.71   | 73.81       | 24.53     | 31.23   |
> | 15000   | 81.33       | 40.39     | 41.56   | 60.21       | 49.95     | 78.62   | 68.20       | 20.94     | 28.70   |
> | 30000   | 77.89       | 36.73     | 38.94   | 52.92       | 46.48     | 81.50   | 59.42       | 14.78     | 22.87   |
>
>
> **Analysis across models:** The context‑length ablation shows **consistent trends**:
> - **NoForget Accuracy** and **Forget Accuracy** show a decreasing trend as the context length increases in three scenarios.
> - **Instructional forgetting** and **dynamic preference** tend to **suffer** (lower IF_SFRR and DP_SFRR) as the context length increases (attention dilution and unstable recency effects).
> - **Subtask revision** tends to **benefit** (higher SR_SFRR) as the context length increases (reduced influence of earlier subtask signals).
>
> We will integrate these cross‑model results and analysis into **Section 4.4** and **Appendix** in the revised version.

---

> ### Author Response · Authors · 2025-11-21
> **Rebuttal (Page 3)**
>
> > Have the authors experimented prompt engineering to improve ICF ability even preliminarily?
>
> We appreciate this valuable suggestion and conducted preliminary experiments comparing whether the prompts used to generate responses include a reminder for the LLM to forget or not.
>
> **Example prompts**
>
> | Scenario                 | NoForget Prompt                                                                                           | Forget Prompt                                                                                                                             |
> |--------------------------|----------------------------------------------------------------------------------------------------------|-------------------------------------------------------------------------------------------------------------------------------------------|
> | Instructional Forgetting  | You are a helpful assistant. You should answer the last question based on history conversation.           | You are a helpful assistant. You should answer the last question based on history conversation. You need to pay attention that if there are some forgotten instructions during the conversation, you should follow these instructions and forget this information. |
> | Subtask Revision          | You are a useful assistant, you can give the answer following my instruction.                            | You are a useful assistant, you can give the answer following my instruction. You should be aware that if similar instructions appear in history, you need to ignore the influence of them and focus on the current instruction. |
> | Dynamic Preference        | You are a useful assistant. You can make the most reasonable choice according to my preferences.          | You are a useful assistant. You can make the most reasonable choice according to my preferences. You should be aware that if the preferences expressed in history are different from the current ones, you need to ignore the different preferences mentioned in history and follow the latest preferences. |
>
> **Results Summary**
>
> Description:
> - three scenarios: instructional forgetting(**IF**)、subtask revision(**SR**)、dynamic preference(**DP**)
> - three metrics: NoForget Accuracy(**NA**)、Forget Accuracy(**FA**)、Selective Forgetting Retention Rate(**SFRR**)
>
> | Model (Setting)               | IF_NA | IF_FA | IF_SFRR | SR_NA | SR_FA | SR_SFRR | DP_NA | DP_FA | DP_SFRR |
> |------------------------------|-------------|-----------|---------|-------------|-----------|---------|-------------|-----------|---------|
> | GPT-O3Mini (Noforget prompt) | 96.48       | 63.08     | 62.88   | 84.51       | 66.08     | 70.93   | 68.62       | 44.90     | 58.18   |
> | GPT-4o (Noforget prompt)     | 95.77       | 52.52     | 52.52   | 81.41       | 62.31     | 74.45   | 86.10       | 44.90     | 50.07   |
> | GPT-5 (Noforget prompt)      | 97.18       | 58.85     | 58.80   | 81.69       | 63.18     | 73.69   | 87.37       | 72.45     | 79.12   |
> | GPT-O3Mini (forget prompt)    | 93.84       | 78.94     | 76.30   | 84.37       | 70.15     | 72.67   | 79.53       | 59.93     | 62.56   |
> | GPT-4o (forget prompt)        | 93.54       | 70.09     | 71.76   | 80.50       | 64.41     | 75.84   | 87.43       | 66.08     | 66.56   |
> | GPT-5 (forget prompt)         | 95.18       | 74.38     | 74.62   | 81.07       | 65.15     | 74.15   | 88.89       | 76.02     | 76.64   |
>
> Our findings are as follows:
> - When the prompts used to generate responses include a reminder for the LLM to forget, **forgetting performance significantly improves** (IF_FA, SR_FA, DP_FA, and their corresponding SFRR metrics) across all tested models. However, **Noforget performance drops a little** in the instructional forgetting scenario(IF_NA metrics), suggesting a potential trade-off between memory retention and in-context forgetting.
> - This suggests that prompt engineering is a promising and practical approach to enhance ICF abilities in current LLMs.
>
>
> We will add these results and discuss the potential of prompt engineering as a effective method for improving ICF in our revised paper.
>
> ---
> We sincerely thank you again for your valuable feedback and hope our responses have addressed your concerns. To ensure the quality of the revised paper, we may need an additional 1–2 days after submitting this rebuttal to finalize the updates.
>
>
> Best regards,

---

### Meta-Review · Area_Chair_SEWt · 2026-01-06

**Summary:**

This paper introduces In-Context Forgetting (ICF) and proposes ICF-Bench, a benchmark evaluating LLMs’ ability to suppress or ignore interfering, outdated, or overridden information during inference without parameter updates. The benchmark contains ~2k multi-turn dialogues across three scenarios with paired NoForget/Forget settings, and proposes SFRR to isolate interference effects from baseline task failures. The problem is timely and relevant to realistic dialogue, privacy control, and preference updates. The benchmark design is clean and systematic, and SFRR is a reasonable metric that mitigates confounds between poor memory and genuine forgetting behavior. Experiments cover a broad set of models; additional rebuttal results substantially strengthen the paper.

The term “forgetting” may be misleading and should be clearly framed as contextual suppression or update-following rather than unlearning. Mechanistic explanations remain high-level and speculative, which is acceptable for a benchmark paper but should be presented cautiously. Context-length results are limited to dialogue-style concatenation and should be scoped accordingly.

**Reviewer Concerns:**

Addressed by the rebuttal: definition of “forgetting”; confound with long-context failures; presentation issues.

Remaining concerns: mechanistic understanding; external generalization.

**Reviewer Scores:**

Reviewer fQrB: explicitly increased score from 2 to 4.
Reviewer XYBR: explicitly increased score from 6 to 8.

---

### Decision · Program_Chairs · 2026-01-26

Accept (Poster)